# Integrated avalanche photodetectors for visible light

Salih Yanikgonul [1,2,6], Victor Leong [1✉], Jun Rong Ong [3✉], Ting Hu[4], Shawn Yohanes Siew [5], Ching Eng Png [3] & Leonid Krivitsky[1]

Integrated photodetectors are essential components of scalable photonics platforms for quantum and classical applications. However, most efforts in the development of such devices to date have been focused on infrared telecommunications wavelengths. Here, we report the first monolithically integrated avalanche photodetector (APD) for visible light. Our devices are based on a doped silicon rib waveguide with a novel end-fire input coupling to a silicon nitride waveguide. We demonstrate a high gain-bandwidth product of 234 ± 25 GHz at 20 V reverse bias measured for 685 nm input light, with a low dark current of 0.12 μA. We also observe open eye diagrams at up to 56 Gbps. This performance is very competitive when benchmarked against other integrated APDs operating in the infrared range. With CMOS-compatible fabrication and integrability with silicon photonic platforms, our devices are attractive for sensing, imaging, communications, and quantum applications at visible wavelengths.

[1] Institute of Materials Research and Engineering, Agency for Science, Technology and Research (A*STAR), Singapore, Singapore. [2] School of Electrical and Electronic Engineering, Nanyang Technological University, Singapore, Singapore. [3] Institute of High Performance Computing, Agency for Science, Technology and Research (A*STAR), Singapore, Singapore. [4] Institute of Microelectronics, Agency for Science, Technology and Research (A*STAR), Singapore, Singapore. [5] Advanced Micro Foundry, Singapore, Singapore. [6] Present address: Advanced Micro Foundry, Singapore, Singapore. ✉email: victor_leong@imre.a-star.edu.sg; ongjr@ihpc.a-star.edu.sg

ntegrated photonics platforms are well poised to meet the growing demands of both classical and quantum applications[1,2]. These platforms can accommodate multiple components on the same chip, including light sources, modulators, and photodetectors[3]. The use of mature CMOS fabrication processes offers scalable manufacturing and deployment of these devices.

On-chip avalanche photodetectors (APDs) are indispensable components of a fully integrated photonics platform. They provide fast detection speeds, high sensitivity down to single-photon levels, and are compatible with waveguide-based designs. The majority of recent research in this area has been geared toward applications in optical communications networks, focusing on operation at infrared telecommunications wavelengths. These devices have been developed on a variety of material platforms, including III–V semiconductors[4], germanium (Ge)[5-11], and Si[12-17].

However, integrated APDs for visible-light detection have yet to be demonstrated. Such devices, if realized, will greatly benefit numerous application areas. For instance, they can lead to miniaturized devices for biomedical imaging[18,19], molecular sensing[20], and underwater imaging[21]. Combined with nanophotonic phased arrays[22,23], they can be applied to visible-light communications[24-26] and bathymetric LIDAR[27]. As APDs do not require cryogenic environments, unlike integrated superconducting photodetectors, they are advantageous for developing scalable systems for quantum information processing, such as the recently demonstrated 100-mode photonic quantum computer Jiuzhang[28]. Integrated visible-light APDs will enable the photonic integration of various quantum systems operating at visible wavelengths, such as trapped ions, color centers in diamond, quantum dots, and 2D materials[29].

A key difficulty in achieving visible-light operation is optical coupling. For integrated APDs, besides the photodetection efficiency of the active APD structure, the coupling of input light from the photonic circuit (especially on-chip waveguides) to the APD is also critical to the device performance. Despite the ubiquity and high performance of free-space APDs for visible-light detection, the coupling of visible light to integrated photodetector structures remains a significant technical challenge. While conventional integrated APDs rely on an interlayer transition from an input waveguide above or below the APD[5,30,31], using the same approach for visible wavelengths would lead to deteriorations in noise and bandwidth performance[15,17]. This is due to the much longer coupling length required to achieve efficient coupling at these wavelengths, resulting in device sizes much larger than what is required for efficient photon absorption. A larger device size decreases the bandwidth due to RC limitation, and also increases dark noise due to the larger active volume.

To date, the shortest operating wavelength among integrated APDs is 850 nm, as demonstrated in devices developed for short-reach data communications[15,17]. To the best of our knowledge, there are no reports of visible-light integrated APDs in the literature.

Here, we present the first demonstration of waveguide-coupled APDs for visible-light detection. To overcome the challenge of input light coupling, we adopt an end-fire coupling configuration between the active APD structure and the input waveguide, which are both fabricated on the same device layer. Our devices are fabricated with CMOS-compatible materials, using silicon nitride ($Si_3N_4$, hereafter denoted as SiN) on a silicon-on-insulator (SOI) platform. The APD is based on a doped Si rib waveguide, while SiN is chosen for the input waveguide for its low propagation loss at visible wavelengths. In contrast to the more conventional interlayer coupling, our end-fire-coupled devices are a novel addition to SiN photonics platforms. We fabricate devices with different device geometries and doping profiles, and characterize their performance, including dark current, gain, dynamic range, bandwidth, and eye diagrams. We then benchmark our results against other recently reported integrated APDs, and show that our devices are indeed very competitive across multiple performance metrics.

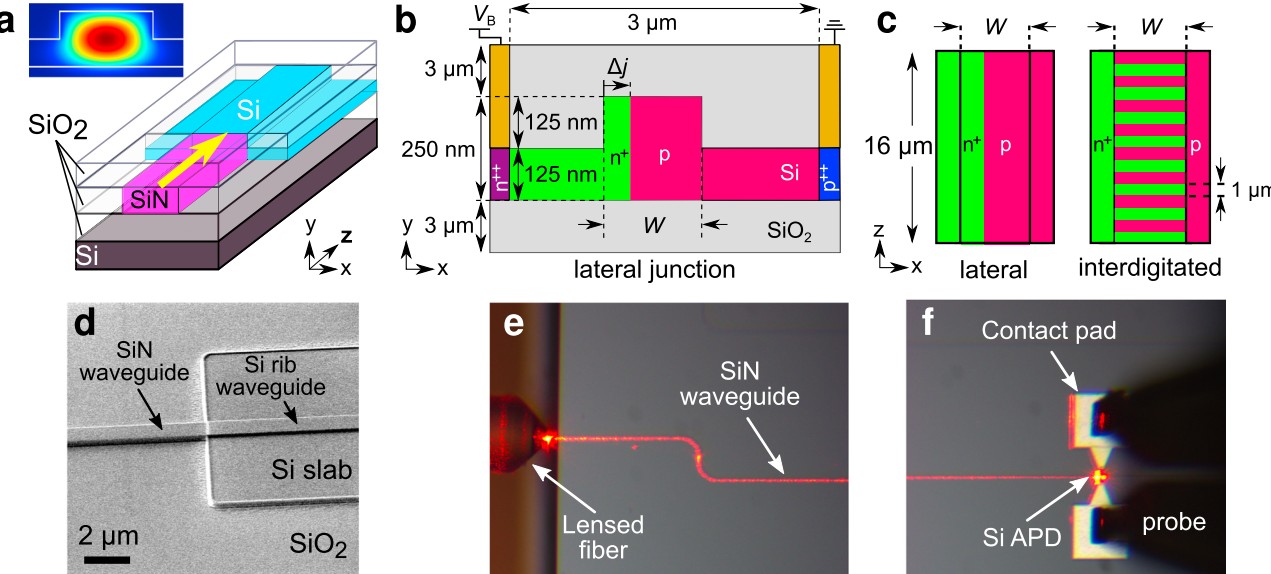

**Fig. 1 Device structure and doping configurations. a** Schematic of the APD device, consisting of a Si rib waveguide end-fire coupled to an input SiN waveguide. The yellow arrow denotes the propagation direction of input light. The inset shows the simulated optical mode in the Si rib waveguide. **b** Cross-sectional view of the Si rib waveguide with a lateral doping profile. The junction placed at a distance $\Delta j$ from the left edge of the waveguide core with a width $W$. A reverse bias voltage $V_B$ is applied via metal contacts deposited on top of heavily doped $p^{++}$ and $n^{++}$ regions. **c** Top view of the Si rib waveguide, showing the lateral and interdigitated doping profiles. **a**–**c** are not drawn to scale. **d** Scanning electron microscope (SEM) image of a fabricated device without the top $SiO_2$ cladding and metal contacts. **e**, **f** Fabricated devices imaged under an optical microscope, showing the lensed fiber coupling and Si APD regions, respectively. The red glow is due to the scattering of the 685-nm input light.

## Results

### Device design

Our device structure is shown in Fig. 1. The primary photodetector structure is a Si rib waveguide of length 16 µm, which has a high absorptivity at visible wavelengths (>96% at 685 nm). Input light is end-fire coupled from an input SiN rectangular waveguide, which allows for low-loss propagation of visible light[32–34]. Both the Si APD and SiN input waveguide have the same width $W$. Two values of $W$, 750 and 900 nm, are explored. The height of both the Si APD and SiN waveguide is fixed at 250 nm, and the Si rib height at 125 nm. The structures are fabricated on a SOI substrate on the same device layer, and are cladded with 3 µm of silicon dioxide (SiO₂) above and below.

To establish electrical connections to the device, metal electrodes are deposited on top of heavily doped p++ and n++ regions at the far ends of the Si slab along the x axis, 3 µm apart.

A careful consideration of the doping profile is required to produce high-performance APDs. Here, we design our APDs to consist of a p–n⁺ diode in two different doping configurations: lateral and interdigitated (see Fig. 1b, c). Both profiles aim to maximize the spatial overlap between the depletion region on the p-doped side and the optical waveguide mode.

The lateral doping profile features a single continuous junction placed asymmetrically along the length of the APD. The design distance between the junction and the n⁺ edge of the waveguide $\Delta j$ is {120, 150} nm for waveguide widths $W$ = {750, 900} nm. We have previously performed simulation studies of this doping profile in Si APDs[35,36]. Though conceptually simple, this profile requires stringent control of the fabrication process, as a small misalignment of the junction will result in a large mismatch between the optical mode and the depletion region.

The alternative design uses an interdigitated profile, which consists of alternating p and n⁺ regions, each 1 µm in length. This design is less sensitive to such misalignment errors, but the increased junction lengths could lead to a higher depletion capacitance and hence limit the bandwidth, as is reported for Si modulators[37,38].

The n⁺ (p) doping concentrations of $1 \times 10^{19}$ ($2 \times 10^{17}$) dopants/cm³ are chosen to ensure that the depletion region covers a large part of the waveguide width in both doping profiles. These doping concentrations are similar to values in other APDs[12,39,40].

In most recent reports on waveguide-based APDs for infrared wavelengths, input light is coupled to the detector via a phase-matched interlayer transition[5,30,31]. However, this is challenging to achieve in a SiN ($n$ = 2.1) to Si ($n$ = 3.8) transition due to the large difference in refractive indices. The increased optical mode confinement at shorter wavelengths means that an efficient interlayer transition would require either long transition lengths, or the narrowing of waveguide dimensions beyond what can be achieved with conventional photolithography[32,33,41,42]. Therefore, we choose to end-fire couple the input SiN waveguide to the Si rib waveguide in the same layer. From our previous analysis of the optical mode overlap between the waveguide modes, we expect a SiN-Si end-fire coupling loss of ≤1 dB per facet[35].

In our fabricated devices, light is coupled into the SiN waveguides via inverse tapers at the edge of the waveguide chip (see Fig. 1e). For both waveguide widths, the inverse tapers are designed to have a taper length of 200 µm and a minimum taper width of 180 nm. The edge-coupled devices are optimized for interfacing with lensed optical fibers; for a focused spot diameter of 2 µm, the expected coupling loss into the SiN waveguide is ~1.5 dB per facet. Detailed characterization of the coupling and propagation losses yield a total insertion loss of 7.1 ± 0.4 dB for our devices (see Supplementary Note 1).

### Current–voltage measurements

We measure the current–voltage (I–V) characteristics of each device up to the breakdown voltage $V_{br}$, with a series of different input optical powers $P_{opt}$ entering the Si waveguide. The values of $P_{opt}$ are reported after accounting for the insertion loss. Here we consider representative results for a $W$ = 900 nm laterally doped device, as shown in Fig. 2a.

From the I–V data we extract the photocurrent $I_{ph} = I_{dev} - I_{dark}$, where $I_{dev}$ and $I_{dark}$ are the measured device current and dark current, respectively. We then obtain the avalanche gain $G$ as the ratio of the photocurrent $I_{ph}$ at bias $V_B$ to that measured at unity gain point of $V_B = 2$ V, where we consider the quantum efficiency to be nearly maximized, and the effects of avalanche gain to be insignificant:

$$G(V_B) = \frac{I_{ph}(V_B)}{I_{ph}(2\,\text{V})} \tag{1}$$

A discussion of how we determined the unity gain point can be found in Supplementary Note 4.

At $V_B > 10$ V, both $I_{dark}$ and $G$ increase dramatically due to avalanche multiplication. In this regime, the power dependence of the device response becomes obvious, with $G$ decreasing for

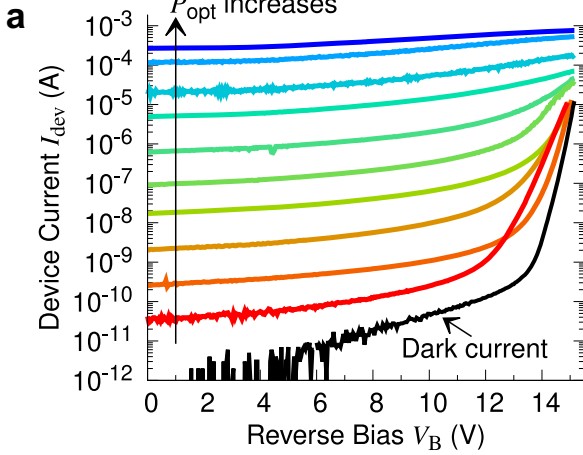

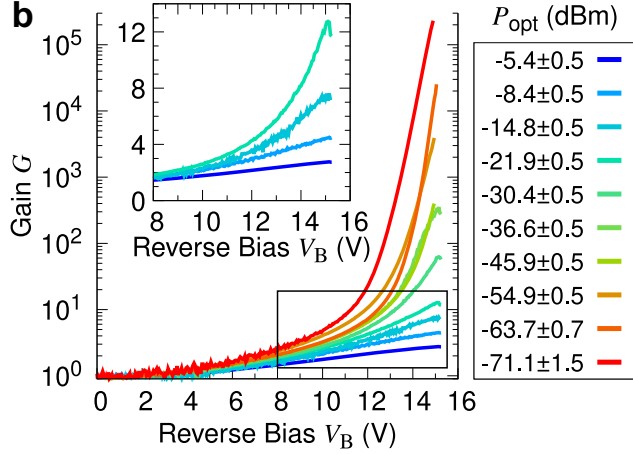

**Fig. 2 DC characteristics of a laterally doped device with width $W$ = 900 nm. a** Current–voltage measurements at different input optical powers $P_{opt}$. The reverse bias voltage $V_B$ is swept till the avalanche breakdown voltage $V_{br} \approx 15.5$ V, where the dark current $I_{dark}$ reaches 10 µA. Each sweep takes a few seconds; prior to each sweep, the device is reset with the application of a forward bias voltage. **b** The avalanche gain $G$ at different $P_{opt}$. The inset is a magnified view of the area marked by the rectangle, showing the curves at larger $P_{opt}$ on a linear scale. Both plots in this figure share the same legend for $P_{opt}$.

higher $P_{opt}$ (see Fig. 2b). This is due to the larger number of multiplied charge carriers causing an increased space charge effect. As a result, the electric field is depressed, leading to saturation of the device current. Thus, while $G \sim 10$ at $V_B = 15$ V for $P_{opt} = -20$ dBm, it rises to $G > 10^5$ for a low input power of $P_{opt} = -71$ dBm. Power-dependent characteristics have also been studied in other APDs[13,43].

As such, we will separately compare the device performance in low-gain and high-gain regimes.

**Performance in the low-gain regime**. In the low-gain regime, the APDs can be operated at small bias voltages suited for applications requiring low power consumption. An important example is

to monitor optical power levels in integrated photonic circuits, which requires low dark current and wide dynamic range with linear response[13,16].

We focus on the primary responsivity $R_p = I_{ph}/P_{opt}$ measured at unity gain, i.e., at $V_B = 2$ V. All device types show linear behavior, with $R_p$ within an overall range of $0.65 \pm 0.18$ A/W over a dynamic range of >50 dB (see Fig. 3a and Table 1). We expect the actual dynamic range to be even larger since we did not explore higher input powers in detail for all devices, and we had not yet observed the device approaching saturation. $R_p$ is slightly higher for $W = 900$ nm devices due to the larger absorption volume of a wider waveguide.

The dark current measurements are shown in Fig. 3b. $I_{dark}$ at $V_B = 2$ V is less than 70 pA for all device types. Laterally doped

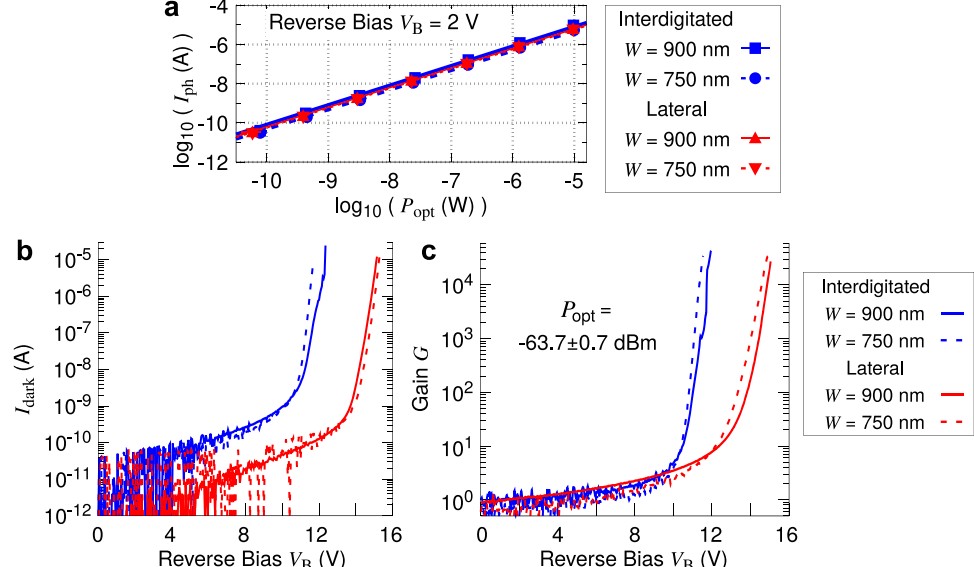

**Fig. 3 Comparison of DC performance for lateral and interdigitated doping profiles with different widths W. a** Photocurrent $I_{ph}$ versus input power $P_{opt}$ at the unity gain point of reverse bias $V_B = 2$ V. Straight lines are linear fits, from which we extract the primary responsivity $R_p$, see Table 1. **b** Dark current $I_{dark}$ measurements at varying $V_B$. **c** Avalanche gain $G$ at varying $V_B$ with a fixed input power $P_{opt} = -63.7 \pm 0.7$ dBm. **b, c** share the same legend on the right.

**Table 1 Benchmarking of device performance with other recent reports of integrated APDs. Results from this work are listed in the top section.**

| Type | $\lambda$ (nm) | $V_B$ (V) | $I_{dark}$ (μA) | $R_p$ (A/W) | Gain | BW (GHz) | GBP (GHz) | Device/Ref. |
|------|------|------|------|------|------|------|------|------|
| Si, LD | 685 | 20 | 0.12 (±1) | 0.83 (±5) | 12.3 (±8) | 19.1 (±8) | 234 (±25) | $W = 900$ nm |
| Si, LD | 685 | 20 | 0.037 (±7) | 0.48 (±2) | 7.7 (±3) | 18.7 (±1) | 144 (±7) | $W = 750$ nm |
| Si, ID | 685 | 18 | 0.31 (±4) | 0.63 (±1) | 2.25 (±6) | 14.8 (±2) | 33 (±1) | $W = 900$ nm |
| Si, ID | 685 | 13 | 0.034 (±3) | 0.56 (±1) | 2.9 (±2) | 16.44 (±8) | 47.4 (±3) | $W = 750$ nm |
| Si | 850 | 14 | 2 | 0.05 | 6 | 16.4 | 98.4[a] | [15] |
| Si | 850 | 20 | 0.016 | 0.071 | 2.2[a] | 13.1 | 28.8[a] | [15] |
| Si | 850 | 12 | 0.0004 | 0.133 | 1.2[a] | 15 | 18[a] | [15] |
| Si | 850 | 20 | 0.001 | 0.24 | 1.3[a] | 4.7 | 6.1[a] | [15] |
| Si | 850 | 20 | 0.075 | 0.2[a] | 1.45[a] | 14 | 20.3[a] | [17] |
| InAs | 1310 | 18.6 | 2000[a] | 0.13[a] | 45 | 5.3[a] | 240 | [4] |
| InAs | 1310 | 15.9 | 0.033 | 0.234 | 20 | 2.06 | 41[a] | [44] |
| Ge/Si | 1310 | 12 | 100 | 0.64 | 11 | 27 | 300 | [10] |
| Ge/Si | 1310 | 18[a] | 0.27 | 0.6[a] | 10 | 36 | 360[a] | [8] |
| Si | 1550 | 9 | 88000[a] | 0.0005[a] | 1080[a] | 26 | 28000 | [16] |
| Ge/Si | 1550 | 13 | 100 | 0.78 | 8.1[a] | 33.8[a] | 274[a] | [10] |
| Ge/Si | 1550 | 6 | 1000[a] | 0.48 | 15 | 18.9 | 284[a] | [9] |
| Ge/Si | 1550 | 10 | 1[a] | 1.25[a] | 17.8[a] | 25 | 445[a] | [11] |

*LD lateral doping, $\lambda$ operating wavelength, $R_p$ primary responsivity, ID interdigitated doping, $V_B$ reverse bias, BW 3 dB bandwidth, W waveguide width, $I_{dark}$ dark current, GBP gain-bandwidth product.*
[a]These values were not explicitly reported, and were inferred from the figures or other values.

devices with $W = 900$ nm exhibit the lowest $I_{dark} \sim 1$ pA (see also Fig. 2a). We note that in the low bias regime ($V_B < 10$ V), laterally doped devices have about an order of magnitude lower $I_{dark}$ than interdigitated devices. This effect has also been previously reported in other waveguide-based photodetectors[13]. There are two likely reasons for the higher dark current in interdigitated devices. First, high peak electric field strengths associated with the corners of the interdigitated regions can lead to a higher dark carrier generation rate[36] (see Supplementary Note 5 for more details on the electric field profiles). Furthermore, the interdigitated devices have a larger depletion volume where dark carriers can undergo avalanche multiplication, compared to their laterally doped counterparts.

**Performance in the high-gain regime.** Figure 3c shows the gain $G$ for different device types at a relatively low input power of $P_{opt} = -63.7 \pm 0.7$ dBm, where the devices exhibit high gain. We see that interdigitated devices have a lower breakdown voltage $V_{br}$ and a slightly steeper rise in $G$ with respect to $V_B$. These effects can likely be attributed to premature breakdown due to high electric fields at the edges of the interdigitated regions. For both doping profiles, we observe no significant dependence of $V_{br}$ on the device width $W$. This is consistent with our previous simulations for laterally doped devices[35].

Applications in integrated photonics typically require low power consumption, thus both $I_{dark}$ and $V_B$ should ideally be low as well[16]. While interdigitated devices achieve similar gain at a lower $V_B$ compared to laterally doped devices, $I_{dark}$ tends to be higher. The optimal choice of doping profile in this regime would then require a more in-depth consideration of the operating requirements.

**High-speed response and bandwidth.** The AC response of the APDs is characterized in the low-gain operation mode, after the device gain has stabilized (see Methods). Figure 4a shows the results of a frequency response measurement for a $W = 900$ nm laterally doped device. The 3 dB bandwidth, which we define with respect to the device response at 1 GHz, is obtained via a smoothing fit to the data points. Figure 4b, c compares the bandwidth and gain-bandwidth product (GBP) of the different device types. At lower reverse bias $V_B$, the bandwidth generally increases with $V_B$ due to a wider depletion region and a lower junction capacitance. However, this effect eventually reaches a limit, beyond which the bandwidth saturates or starts to decrease, due to the device response being limited by avalanche buildup times at large gain[16]; this occurs at $V_B \sim 12$ V in our devices.

We find that the bandwidth is indeed lower in interdigitated devices, as expected from the higher capacitance due to its doping profile. Another potential contributing factor is that a larger proportion of photo-generated charge carriers in interdigitated devices is created in n$^+$-doped regions where the electric field is low, leading to slower carrier diffusion and hence slower device response (see Supplementary Note 5).

A detailed comparison of the best GBP performance for each device is shown in Table 1. The highest observed GBP is $234 \pm 25$ GHz for the $W = 900$ nm laterally doped device, at a reverse bias of $V_B = 20$ V. Although its $W = 750$ nm version has a lower maximum GBP, it also has a much lower dark current, as well as higher 3 dB bandwidths of up to 30 GHz at lower $V_B$. As such, the optimal choice of device parameters might also depend on the specific application and operating conditions.

To demonstrate the performance of our devices in communications systems, we measured eye diagrams of the different device types (see Fig. 5). Lateral devices show open eyes at data rates of up to 56 Gbps at $V_B = 20$ V, where the maximum GBP is observed. We note that these devices can potentially support even

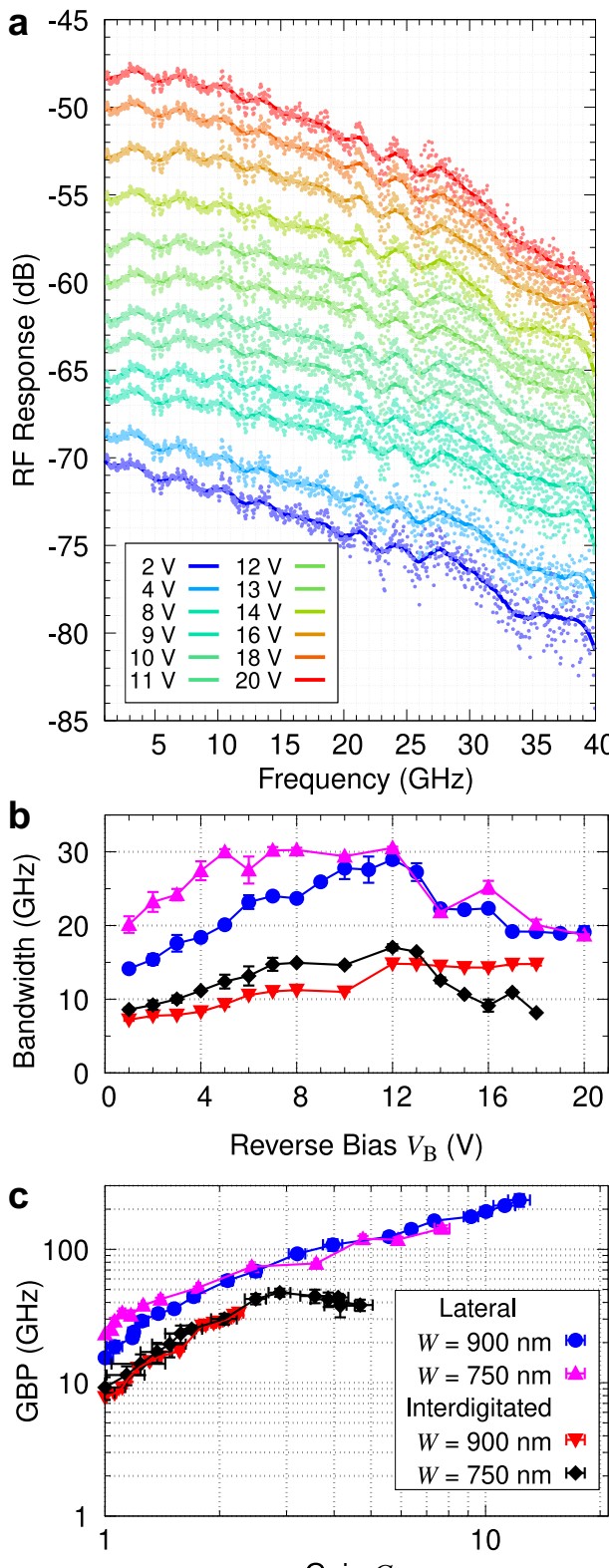

**Fig. 4 Optical–electrical bandwidth measurements.** An input power of $P_{opt} = -24.5$ dBm is used throughout. **a** Frequency response of a $W = 900$ nm laterally doped device at various bias voltages $V_B$. The 3 dB bandwidth is obtained from a smoothing fit to the data points (see Methods). **b**, **c** The 3 dB bandwidth and gain-bandwidth product (GBP), respectively, for different device types. Both plots share the same legend shown in **c**. Each data point and error bar in both plots represent the mean and standard deviation, respectively, of several measurements.

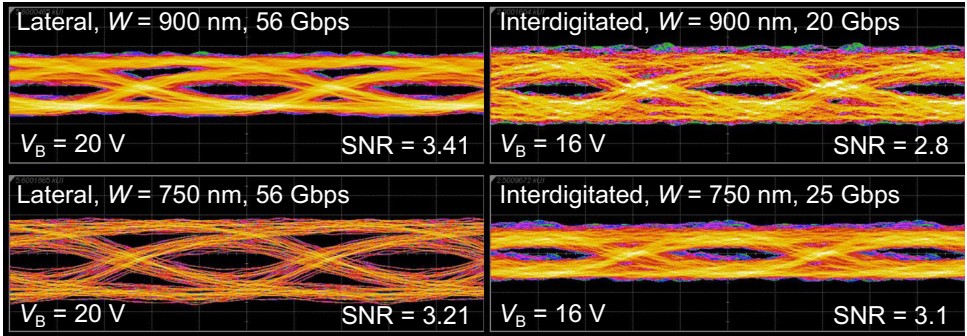

**Fig. 5 Measured eye diagrams for the different device types.** Lateral devices show open eyes at data rates of up to 56 Gbps at $V_B = 20$ V, where the maximum GBP is observed. The results for interdigitated devices are obtained at the highest data rate where open eyes could be measured for each device. The signal-to-noise ratio (SNR) is obtained from the sampling oscilloscope.

higher data rates at lower bias, where the 3 dB bandwidth is higher, but our setup is not capable of generating faster bit patterns. Interdigitated devices show open eyes only at lower data rates, with the $W = 750$ nm devices performing slightly better (25 Gbps) than $W = 900$ nm ones (20 Gbps). This is attributed to the narrower devices having a larger gain ($G \sim 4$ at $V_B = 16$ V, while $G \sim 2$ for the wider devices), despite having a slightly lower 3 dB bandwidth as seen in Fig. 4b. We note that all devices can be operated at lower data rates with a higher signal-to-noise ratio (SNR); additional eye diagrams showing this can be found in Supplementary Note 3.

We note that the high-speed performance of our devices can be adversely affected by factors such as the size of contact pads, which could be further reduced or removed altogether in future large-scale integration with a photonics platform.

## Discussion

Table 1 shows the benchmarking of our device performance with other recent reports of integrated APDs. Where possible, we report the performance of each device at the operating conditions where the maximum GBP is observed. For all the devices we benchmark against, we omit the uncertainty values, as only some of the literature reports include this information. We note that in the literature, the primary responsivity and unity gain are reported at varying bias voltages; where not explicitly defined, we have extracted the relevant values at a bias of $V_B = 1$ V, following ref. [16]. For our devices, we also note that the values for the dark current $I_{dark}$ in Table 1 are measured in a different regime compared to Fig. 3b, where the reset procedure is used (see Supplementary Note 7).

Our best-performing device is the $W = 900$ nm laterally doped APD, with a GBP of 234 GHz. Compared to other contemporary devices, this APD shows a strong, balanced performance in the performance metrics of dark current $I_{dark}$, primary responsivity $R_p$, gain, and bandwidth. With the exception of ref. [16] which has a very high operating $I_{dark}$ of 88 mA, the 234 GHz GBP of our APD is also comparable to the highest reported values of a few hundred GHz. Yet, our APD also exhibits a much lower $I_{dark}$ of 0.12 μA at the operating bias $V_B$ than other high-GBP devices; this would lead to decreased noise and power consumption.

These observations show that our devices are competitive, and are well-suited for visible-light applications requiring high bandwidth and high sensitivity.

In conclusion, we have reported the first fabrication and characterization of waveguide-integrated Si APDs for visible light (685 nm). Our devices feature a small device footprint and are fabricated with a CMOS-compatible process. At a reverse bias of $V_B = 2$ V, a laterally doped APD of 900 nm width exhibited a highest primary

responsivity of $0.83 \pm 0.05$ A/W over a dynamic range of >50 dB, with dark current of ~1 pA. At higher $V_B$, laterally doped devices exhibit superior bandwidth, with a highest 3 dB bandwidth of 30.5 $\pm 0.2$ GHz, and a highest GBP of $234 \pm 25$ GHz. APDs with an interdigitated doping profile require a lower bias to attain the same DC gain than lateral ones, but have a higher dark current. Our devices perform strongly compared to other state-of-the-art integrated APDs operating at other wavelengths.

The addition of integrated visible-light APDs to the component toolbox of SiN photonics opens up many application possibilities, and greatly expands the versatility of silicon photonics platforms[32–34]. There is potential for further design optimizations, such as alternative doping profiles[16] which may enhance the APD gain and reduce the working bias. Future work will also explore the operation of these devices in the Geiger mode for single-photon counting, which will play an important role in the development of integrated quantum photonics platforms, and for interfacing with single-photon sources operating at visible wavelengths.

## Methods

**Device fabrication**. The devices were fabricated at Advanced Micro Foundry and the Institute of Microelectronics, A*STAR. The main fabrication steps of the device are as follows: we start from an 8-inch SOI wafer, with 220 nm Si and 3 μm buried oxide (BOX) layers. An epitaxy of Si (30 nm) tops up the total Si thickness to 250 nm. We then form the Si slab using 248 nm KrF deep-UV lithography and inductively coupled plasma (ICP) etch.

We then deposit a 450-nm thick SiN layer using low-pressure chemical vapor deposition, and reduce it to the same height as the Si slab (250 nm) with chemico-mechanical polishing followed by wet etch. Next, we use lithography and ICP etch with an oxide hard mask to pattern the SiN waveguides and then the Si rib waveguides (125 nm etch into the Si slab) in subsequent steps. The image shown in Fig. 1d was taken after removing the oxide hard mask.

We perform implantation of the p and $n^+$ regions along the Si rib waveguide, followed by the $p^{++}$ and $n^{++}$ ohmic contact regions with a subsequent rapid thermal anneal at 1030 °C for 5 s. We then deposit 3 μm of oxide as the top cladding, followed by the opening of contact holes. Finally, we deposit and pattern aluminum to form the contact pads.

Over the course of our measurements, we have tested several tens of devices with different device parameters picked from various locations across an 8" wafer. All tested devices show repeatable results, and we did not observe a single failed device. This indicates that the fabrication is robust and has a high device yield.

**Characterization setup**. We test the fabricated devices at room temperature using a custom-built light-tight probe station (see Supplementary Note 2 for the setup schematic). We establish electrical connections via $100 \times 100$ μm contact pads on the chip surface using electrical probes (see Fig. 1f). We use a 685-nm continuous wave diode laser (Thorlabs LP685-SF15) as the optical source. The laser light is coupled to the SiN waveguide using single-mode tapered lensed fibers (OZ Optics TSMJ-3U-633-4/125-1-30-2-9-1, 2 μm spot diameter).

We maintain a horizontal input polarization, which couples to the fundamental TE mode of the SiN waveguide. Although different input polarizations could lead to some variations in the coupling and propagation losses, the APD response itself is not expected to exhibit any significant polarization dependence.

**Electro-optic characterization**. For I–V measurements, the reverse bias voltage $V_B$ is swept from 0 V to the avalanche breakdown voltage $V_{br}$ over a few seconds. We define $V_{br}$ here as the voltage where the dark current $I_{dark}$ (i.e., without input light) reaches 10 µA; this definition follows other reports of APDs in the literature[45,46]. We note here that $V_{br}$ drifts with time in our devices; as such, to ensure consistent results, it is necessary to reset the device with the application of a forward bias voltage prior to each sweep. More details regarding the drift behavior are discussed in Supplementary Notes 6 and 7.

For bandwidth measurements and eye diagram measurements, the device gain is first stabilized by continuously applying a reverse bias over ~30 min; this is necessary due to the drift behavior. The 685-nm input light is modulated with an RF signal using a 40 GHz electro-optic modulator (EOM, Eospace AZ-AV5-40-PFA-PFA-700). The EOM is operated at 65 °C to mitigate the photorefractive effects caused by high optical input powers. The EOM is maintained at its half transmission point, i.e., the DC bias is adjusted such that the EOM output power is at 50% of its maximum value, before RF modulation is added.

The frequency response is measured with an Agilent E8363C network analyzer, which generates the RF signal for the EOM and measures the APD response. For all devices, we use an input power of $P_{opt} = -24.5$ dBm. The measured data are corrected for the electro-optic S21 response of the EOM, and smoothed with a Savitzky–Golay filter with a third-order polynomial fit. The 3 dB bandwidth is extracted from the fit function.

For eye diagram measurements, a bit pattern generator (SHF 12104 A together with Anritsu MG3693C) is used to generate non-return-to-zero on-off-keying patterns (NRZ-OOK) with pseudorandom binary sequences (PRBS) of length $2^7 - 1$. These patterns are then amplified (Centellax OA4MVM3) and used to modulate the RF signal driving the EOM. Reference eye diagrams of the EOM output are shown in Supplementary Note 3. The eye diagrams are measured with a sampling oscilloscope (Keysight 86100D with 86116C module). An additional remote sampling head (Keysight N1046A) was used at 56 Gbps to obtain a clearer signal.

## Data availability

Source data for Figs. 2–4 and Supplementary Figs. 1, 3b, 4, 5b, 5c, 7, and 8 are provided with the paper. All other data supporting the findings of this study are available from the authors upon reasonable request.

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

## Acknowledgements

S.Y. would like to acknowledge the support of the Singapore International Graduate Award (SINGA). The authors thank Mingbin Yu and Shiyang Zhu for their inputs at the initial stage of the project. We acknowledge the support of the National Research Foundation, Singapore (Grant No. NRF-CRP14-2014-04).

## Author contributions

S.Y., S.Y.S., and V.L. designed and performed the experiments. J.R.O. led the device design and simulation work. T.H. contributed to the device fabrication. L.K. conceived the idea. C.E.P. and L.K. supervised the work. All co-authors contributed to writing and proofreading the manuscript.

## Competing interests

The authors declare no competing interests.
