## [Peer Review File · Nature Communications]

REVIEWER COMMENTS

Reviewer #1 (Remarks to the Author):

This is an interesting paper. I confess that I thought every Si APD paper that could be written had been decades ago. However, it appears that the authors are correct that integrated Si APDs that operate in the visible have not been widely studied. The authors list some potential applications, most of which I agree with, however, while a visible LIDAR paper may have been written, LIDAR will be developed at eye-safe wavelengths. Also, it is not clear how visible light communications, which will be primarily free space, would utilize a waveguide photodiode. Nevertheless, the authors have fabricated and thoroughly characterized a waveguide Si APD. My primary concern with this paper is the authors' determination of gain. They have measured the photoresponse at 685 nm and used the 1 V bias as the unity gain point. However, it is clear from the photoresponse curves that there is a bias-dependent photoresponse and it is not reasonable to attribute it entirely to avalanche gain. For example, the signal at 4 V is higher than that at 1 V but the gain at 4 V, given the electric field should still be unity. It would help to know the quantum efficiency at 1 V to determine if that corresponds to the expected photocurrent for total absorption. I suspect it does not because the authors note in describing the frequency response that the depletion region still increases with bias at even higher bias. One could respond, "so what, the gain might be off by 50% or at most a factor of 2 or 3". For maximum gains of 10,000, those corrections seem minor. However, that can significantly impact the estimated gain-bandwidth. Therefore, I request that the authors actually provide the quantum efficiency at 1 V and at 4 V and compute the gain expected based on the published ionization coefficients.

Reviewer #2 (Remarks to the Author):

I split the review between novelty and presented evidence

Novelty

The authors present a waveguide integrated avalanche photodetector. The actual detector is based on a silicon rib-waveguide with pn junction (lateral or interdigitated). The detector is optically fed by a silicon nitride waveguide.

The first important feature for comparison with other silicon photodetectors is the wavelength range where the detector is operating, because this will determine the light detection mechanism governing the device's optoelectronic conversion function. The authors characterized their device at 685nm (visible light with 1.81eV photon energy). This is indeed a novelty. However, similar photodetector operation at 850nm wavelength was previously demonstrated (ref. 9 & 11). The wavelength of 850nm corresponds to 1.46eV photon energy. The small difference in photon energy does not alter the absorption depth significantly. The photoelectric response of the new device and the previous devices in ref. 9 & 11 is governed by interband transitions. Therefore, the claim of novelty regarding a visible light avalanche detector is correct, but not of major significance.

In terms of device structure, very similar waveguide integrated silicon photodetectors were recently demonstrated by another group, and are referenced by the authors (ref. 11).

As for responsivity, the authors of ref. 11 use a coupling scheme (inverse taper) between the silicon nitride waveguide and the silicon waveguide detector which is different from the present work (butt coupling). However, it is known from germanium waveguide detectors that high coupling efficiency to the absorber can be achieved for both schemes. A more important factor than the coupling scheme is the way of coupling the light to the chip. The authors of ref. 11 used waveguide gratings for fiber-chip coupling. The reported responsivity in ref. 11 was a factor 2-3 smaller than the responsivity of the here presented devices, but probably - at least partially - this should not be attributed to the detector itself but to the way of fiber chip coupling.

Regarding the gain-bandwidth product (GBP), the new device does indeed achieve the highest GBP of all published waveguide detectors in the near-infrared. However, the new device does not show orders of magnitude of improvement.

Regarding dark current, similar dark current values are achieved in ref. 9, and should be cited for comparison in Table 1.

Evidence

The evidence to prove novelty is gathered in the benchmark table 1. In my opinion, this table does not fulfill the standards of a scientific article. A highly significant performance figure for avalanche photodetectors is dark current. As dark current scales with device size, dark current should also be normalized and the figure dark current density should be provided. In addition, the comparison with small gap semiconductors does not make sense, as they are intended for a different wavelength range. Values have been selectively chosen (e.g. dark current for one specific device from ref. 9), which makes it harder to put the new device performance into the context of what other groups achieved.

I do not believe the presented work is of extreme importance to other scientist working in the field.

Reviewer #3 (Remarks to the Author):

Review of "Integrated Avalanche Photodetectors for Visible Light" by SALIH YANIKGONUL et al.

The authors claimed to report the first monolithically integrated APD for visible light at 685 nm. The demonstrated device uses an end-fire coupling to a Si₃N₄ waveguide. The gain-bandwidth product is 216 at a 20-V reverse bias with a 12-GHz bandwidth. In Tab. 1 the authors compare this APD with other devices at the telecom wavelengths, and the overall performances look good. The fabrication process is CMOS compatible, making the technology attractive for visible light APDs.

The manuscript can be considered for publication if the following issues are addressed:

1. The design and the fabrication of the devices are not special; I am wondering why the visible light APD was not demonstrated previously? What were the significant technical challenges that prevented the researchers in the APD field from exploiting the visible spectral range?
2. While the results of the visible light APD look good, what are the key enabling techniques? After reading through the manuscript, the readers may not be able to easily catch the breakthrough techniques.
3. Can the authors comment on the design tolerance and the fabrication robustness of the device? and the typical yield?
4. The authors claimed that "However, using the same approach for visible wavelengths would lead to deteriorations in noise and bandwidth performance, due to the much longer coupling length required to achieve efficient coupling." – I don't quite understand this, can the authors elaborate a bit?
5. It would be more convincing to see an eye diagram of a detected high-speed signal, in addition to the frequency responses provided in the manuscript.
6. If the APD is intended for applications including both conventional and quantum systems, it would be helpful to show some (even preliminary) results on the single photon detection, to evaluate its sensitivity performance for quantum detection. But this is optional.
7. What fiber was used to carry the 685-nm light? What laser source did the authors use?
8. It would be helpful to add some simulation results in Fig. 1a to show the optical fields.

If the above questions could be well addressed, I would recommend publication of this manuscript.

Response to Reviewers' comments

Reviewer #1

This is an interesting paper. I confess that I thought every Si APD paper that could be written had been decades ago. However, it appears that the authors are correct that integrated Si APDs that operate in the visible have not been widely studied. The authors list some potential applications, most of which I agree with, however, while a visible LIDAR paper may have been written, LIDAR will be developed at eye-safe wavelengths. Also, it is not clear how visible light communications, which will be primarily free space, would utilize a waveguide photodiode.

Response: We agree with the Reviewer that land-based LIDAR applications will largely be developed at eye-safe wavelengths. We have updated our LIDAR reference (ref 27) to specifically refer to bathymetric LIDAR, which relies on green wavelengths to penetrate water bodies.

We thank the Reviewer for the comment that the applicability of waveguide photodiodes to free-space communications is not clear. Integrated APDs can be combined with nanophotonic phased arrays for LIDAR and free-space communications applications. We have now clarified this in our introduction, and included additional references, see refs 22, 23.

In addition, we have also highlighted in the introduction the recent demonstration of quantum advantage with a photonic quantum computer by the Hefei group led by Jian-Wei Pan (Science **370**, 1460). The work emphasises the importance of integrated photonic platforms in the future scaling of this technology. Our contribution towards integrated APDs has a direct relevance to the realization of such a platform.

Nevertheless, the authors have fabricated and thoroughly characterized a waveguide Si APD. My primary concern with this paper is the authors' determination of gain. They have measured the photoresponse at 685 nm and used the 1 V bias as the unity gain point. However, it is clear from the photoresponse curves that there is a bias-dependent photoresponse and it is not reasonable to attribute it entirely to avalanche gain. For example, the signal at 4 V is higher than that at 1 V but the gain at 4 V, given the electric field should still be unity. It would help to know the quantum efficiency at 1 V to determine if that corresponds to the expected photocurrent for total absorption. I suspect it does not because the authors note in describing the frequency response that the depletion region still increases with bias at even higher bias. One could respond, "so what, the gain might be off by 50% or at most a factor of 2 or 3". For maximum gains of 10,000, those corrections seem minor. However, that can significantly impact the estimated gain-bandwidth. Therefore, I request that the authors actually provide the quantum efficiency at 1 V and at 4 V and compute the gain expected based on the published ionization coefficients.

Response: We agree with the Reviewer that a more rigorous determination of the unity gain point is required. We now empirically estimate the unity gain point to be just under

2V by determining the point at which the second derivative of the measured photocurrent with respect to the bias voltage is zero. In addition, our simulation results (see Supplementary Note 4) indicate that even at ~2V bias, the gain already starts to increase and the quantum efficiency has saturated (>0.95). Thus, we now set 2V bias as the unity gain point for all devices, and have updated all relevant figures and results accordingly in the manuscript.

We invite the Reviewer to refer to Supplementary Note 4 of our updated Supplementary Information for a more in-depth discussion of the determination of the unity gain point.

Reviewer #2

I split the review between novelty and presented evidence

Novelty

The authors present a waveguide integrated avalanche photodetector. The actual detector is based on a silicon rib-waveguide with pn junction (lateral or interdigitated). The detector is optically fed by a silicon nitride waveguide.

The first important feature for comparison with other silicon photodetectors is the wavelength range where the detector is operating, because this will determine the light detection mechanism governing the device's optoelectronic conversion function. The authors characterized their device at 685nm (visible light with 1.81eV photon energy). This is indeed a novelty. However, similar photodetector operation at 850nm wavelength was previously demonstrated (ref. 9 & 11). The wavelength of 850nm corresponds to 1.46eV photon energy. The small difference in photon energy does not alter the absorption depth significantly. The photoelectric response of the new device and the previous devices in ref. 9 & 11 is governed by interband transitions. Therefore, the claim of novelty regarding a visible light avalanche detector is correct, but not of major significance.

In terms of device structure, very similar waveguide integrated silicon photodetectors were recently demonstrated by another group, and are referenced by the authors (ref. 11).

Response: The devices demonstrated in ref. 11 (Chatterjee et al., Optics Letters **44**, 1682 (2019)) do indeed have a similar active APD structure based on a Si rib waveguide. However, in the design of integrated APDs, beyond the optoelectronic conversion as the Reviewer had mentioned, it is also important to consider the efficiency of input light coupling from a photonic circuit. Although we agree with the Reviewer that the small difference in photon energy does not significantly alter the absorption depth, it requires a completely different approach to input light coupling to ensure overall device performance.

Beyond extending the operation of integrated APDs to visible wavelengths, the novelty in our work also involves the achievement of an efficient butt-coupling configuration on a SiN photonics platform.

We have edited the introduction and device design sections to explain these points more clearly.

We also further elaborate on the butt-coupling configuration below.

As for responsivity, the authors of ref. 11 use a coupling scheme (inverse taper) between the silicon nitride waveguide and the silicon waveguide detector which is different from the present work (butt coupling). However, it is known from germanium waveguide detectors that high coupling efficiency to the absorber can be achieved for both schemes. A more important factor than the coupling scheme is the way of coupling the light to the chip. The authors of ref. 11 used waveguide gratings for fiber-chip coupling. The reported responsivity in ref. 11 was a factor 2-3 smaller than the responsivity of the here presented devices, but probably - at least partially - this should not be attributed to the detector itself but to the way of fiber chip coupling.

Response: We agree with the Reviewer that efficient coupling of IR light from waveguides to the absorber can be achieved via an inverse taper. However, this approach is not suitable for shorter wavelengths, as a longer coupling length is required (on the order of hundreds of microns), which leads to poorer noise and bandwidth performance. In fact, the authors of ref. 11 have also noted the tradeoff between coupling efficiency and bandwidth performance. Although they report a simulated coupling efficiency of 72% from the SiN waveguide to the Si absorber, no experimental values were presented. As a comparison, we note that in our butt-coupled devices, the simulated SiN-to-Si coupling loss is ≤ 1 dB (mentioned in the main text).

Regarding the fiber-to-chip coupling, ref. 11 reports that the responsivity is “*measured from the DC photocurrent and power coupled into the waveguide*”, and also mentioned the use of reference SiN waveguides. As such, the reported responsivity values in ref 11 cannot be attributed to the fiber-to-chip coupling, as the associated loss has already been accounted for.

Regarding the gain-bandwidth product (GBP), the new device does indeed achieve the highest GBP of all published waveguide detectors in the near-infrared. However, the new device does not show orders of magnitude of improvement.

Response: We thank the Reviewer for recognising that we have achieved a higher GBP than the published near-infrared waveguide detectors. This shows that our devices show strong performance and are indeed very competitive.

Regarding dark current, similar dark current values are achieved in ref. 9, and should be cited for comparison in Table 1.

Response: We had previously included only the results from ref. 9 (Fard et al, Optics Express **25**, 5107 (2017)) that the authors described as the “optimized device”. As the Reviewer has requested, results from the other devices in ref. 9 have now been included in the benchmarking table.

Evidence

The evidence to prove novelty is gathered in the benchmark table 1. In my opinion, this table does not fulfill the standards of a scientific article. A highly significant performance figure for avalanche photodetectors is dark current. As dark current scales with device size, dark current should also be normalized and the figure dark current density should be provided. In addition, the comparison with small gap semiconductors does not make sense, as they are intended for a different wavelength range. Values have been selectively chosen (e.g. dark current for one specific device from ref. 9), which makes it harder to put the new device performance into the context of what other groups achieved.

Response: We believe that the total dark current, not dark current density, ultimately determines the performance of any practical device. Therefore, we have designed the device to have a small size in order to keep the overall dark current low. We also note that all the reports we have benchmarked our devices against also report the dark current, not the dark current density, see References listed in Table 1 of the manuscript.

To the best of our knowledge, our work is the first demonstration of integrated avalanche photodetectors in the visible wavelength range. This is why it is only possible to make a comparison with devices operating in a different wavelength range.

As mentioned above, the other values from ref. 9 have been added to the table.

I do not believe the presented work is of extreme importance to other scientist working in the field.

Response: We hope that our response would convince Reviewer #2 on the suitability of our work for the publication in Nature Communications. Furthermore, we also want to point his/her attention to the recent demonstration of quantum advantage with a photonic quantum computer by Jian-Wei Pan’s group (Science **370**, 1460). The work emphasises the importance of integrated photonic platforms in future scaling of this technology. Our contribution towards integrated APDs for visible light has a direct relevance to the realization of such a platform.

Reviewer #3

The authors claimed to report the first monolithically integrated APD for visible light at 685 nm. The demonstrated device uses an end-fire coupling to a Si₃N₄ waveguide. The gain-bandwidth product is 216 at a 20-V reverse bias with a 12-GHz bandwidth. In Tab. 1 the authors compare this APD with other devices at the telecom wavelengths, and the overall performances look good. The fabrication process is CMOS compatible, making the technology attractive for visible light APDs.

The manuscript can be considered for publication if the following issues are addressed:

1. The design and the fabrication of the devices are not special; I am wondering why the visible light APD was not demonstrated previously? What were the significant technical challenges that prevented the researchers in the APD field from exploiting the visible spectral range?
2. While the results of the visible light APD look good, what are the key enabling techniques? After reading through the manuscript, the readers may not be able to easily catch the breakthrough techniques.

Response for #1 and #2: Integrated APDs require both an efficient photoelectric response of the active detector structure, as well as an efficient coupling of input light to the APD from the photonic circuit. The coupling of visible light to integrated APDs has remained a significant challenge, due to the drawbacks of the conventional interlayer transition coupling scheme at visible wavelengths. This is further elaborated in our response to point #4.

The key enabling technique is the achievement of an end-fire coupling configuration for coupling input light from the SiN waveguide to the Si APD. This is a novel addition to SiN photonics platforms, and allows us to achieve efficient input coupling while avoiding the drawbacks of an interlayer transition.

Based on the feedback of the Reviewer, we have edited our introduction section to explain these points more clearly.

3. Can the authors comment on the design tolerance and the fabrication robustness of the device? and the typical yield?

Response: Over the course of our measurements, we have tested several tens of devices with different device parameters, picked from various locations across an 8" wafer. We did not observe a single failed device among the ones we have tested. All the tested devices show repeatable results. The detailed parameter sensitivity studies are conducted in our earlier simulation papers, see Refs 35 and 36 of the updated manuscript. They show decent robustness of the devices to fabrication tolerances.

While we do not have the capacity to perform mass wafer-scale testing, our results indicate that the fabrication is robust and has a high device yield. We have added a corresponding note in the revised manuscript.

4. The authors claimed that “However, using the same approach for visible wavelengths would lead to deteriorations in noise and bandwidth performance, due to the much longer coupling length required to achieve efficient coupling.” – I don’t quite understand this, can the authors elaborate a bit?

Response: At shorter wavelengths, the optical mode confinement within the waveguides is stronger. To achieve efficient interlayer coupling at visible wavelengths, the waveguide dimensions would have to be narrowed beyond what can be achieved with conventional photolithography. Alternatively, we can achieve efficient coupling using a very long (on the order of a few hundred microns) coupling length compared to IR devices. However, this would then require a much larger device size, which comes with significant drawbacks, such as higher dark noise and lower bandwidth. Indeed, the dark noise scales with active device volume, thus larger devices have worse noise performance. In addition, larger devices have lower bandwidths as the bandwidth becomes RC-limited.

The introduction and device design sections have also been updated to explain this more clearly.

5. It would be more convincing to see an eye diagram of a detected high-speed signal, in addition to the frequency responses provided in the manuscript.

Response: We thank the Reviewer for this suggestion. We have measured open eye diagrams at up to 56Gbps. These are now included in the main text of the manuscript. The details of the experimental setup are outlined in the Methods section of the revised manuscript.

6. If the APD is intended for applications including both conventional and quantum systems, it would be helpful to show some (even preliminary) results on the single photon detection, to evaluate its sensitivity performance for quantum detection. But this is optional.

Response: We agree that it would be helpful to show preliminary results on single photon detection. Unfortunately, we are not ready to show these results.

7. What fiber was used to carry the 685-nm light? What laser source did the authors use?

Response: We have stated in the Methods section that we use a 685nm continuous wave diode laser, carried by single-mode tapered lensed fibers from OZ optics. We now further specify the model numbers for both the diode laser (Thorlabs LP685-SF15) and lensed fiber (OZ optics TSMJ-3U-633-4/125-1-30-2-9-1) in the text.

8. It would be helpful to add some simulation results in Fig. 1a to show the optical fields.

Response: We thank the Reviewer for this suggestion. Optical field simulations have been added to Fig. 1a.

If the above questions could be well addressed, I would recommend publication of this manuscript.

REVIEWERS' COMMENTS

Reviewer #1 (Remarks to the Author):

I think the authors have adequately answered the reviewers' comments and that the paper is now acceptable for publication.

Reviewer #3 (Remarks to the Author):

The authors have well addressed all my comments; now the manuscript is ready to move forward towards publication.

REVIEWERS' COMMENTS

Reviewer #1 (Remarks to the Author):

I think the authors have adequately answered the reviewers' comments and that the paper is now acceptable for publication.

Reviewer #3 (Remarks to the Author):

The authors have well addressed all my comments; now the manuscript is ready to move forward towards publication.

Reply: We thank the reviewers for their comments and for reviewing our revised manuscripts. The comments have really helped us to improve our manuscript quality.